# Female Gender Is Associated with an Increased Left Ventricular Ejection Fraction Recovery in Patients with Heart Failure with Reduced Ejection Fraction

**DOI:** 10.3390/medsci10020021

**Published:** 2022-04-02

**Authors:** Jakrin Kewcharoen, Angkawipa Trongtorsak, Sittinun Thangjui, Chanavuth Kanitsoraphan, Narut Prasitlumkum

**Affiliations:** 1Division of Cardiology, Loma Linda University Health, Loma Linda, CA 92354, USA; 2AMITA Health Saint Francis Hospital, Chicago, IL 60202, USA; angkawipa.t@gmail.com; 3Internal Medicine Residency Program, Bassett Healthcare Network, New York, NY 13326, USA; s.thangjui@gmail.com; 4University of Hawaii Internal Medicine Residency Program, Honolulu, HI 96813, USA; winkub@gmail.com; 5Department of Cardiology, University of California Riverside, Riverside, CA 92521, USA

**Keywords:** female, LVEF recovery, heart failure, recovered LVEF, reduced LVEF

## Abstract

We performed a systematic review and meta-analysis to evaluate the association whether the female gender was associated with an increased chance of left ventricular ejection fraction (LVEF) in patients with heart failure with reduced ejection fraction (HFrEF). We searched the databases of MEDLINE and EMBASE from inception to 18 January 2022. Included studies were published studies evaluating or reporting characteristics of patients with HF with recovered LVEF. Data from each study were combined using a random-effects model, the generic inverse variance method of DerSimonian and Laird, to calculate odd ratios (OR) and 95% confidence intervals (CI). Eighteen studies were included in the analysis with a total of 12,270 patients (28.2% female). Female gender was associated with an increased chance of LVEF recovery (pooled OR = 1.50, 95% CI = 1.21–1.86, *p*-value < 0.001, I^2^ = 74.5%). In our subgroup analysis, female gender was associated with an increased chance of LVEF recovery when defined as LVEF > 50% (pooled OR = 1.78, 95% CI = 1.45–2.18, *p*-value < 0.001, I^2^ = 0.0%), and LVEF > 40–45% (pooled OR = 1.45, 95% CI = 1.09–1.91, *p*-value = 0.009, I^2^ = 79.2%), but not in LVEF > 35 (OR = 2.18, 95% CI = 0.94–5.05, *p*-value = 0.06). Our meta-analysis demonstrated that the female gender is associated with an increased chance of LVEF recovery. This association was not statistically significant in the subgroup that defined LVEF recovery as LVEF > 35%.

## 1. Introduction

Heart failure (HF) is a common cardiac disease that contributes to more than 1 million hospitalizations annually in the United States [1]. The severity of this cardiac condition can be stratified by left ventricular ejection fraction (LVEF). The European Society of Cardiology (ESC) 2016 guidelines has categorized HF into three different groups based on different prognoses and approaches to management: heart failure with reduced EF (HFrEF) with LVEF < 40%; heart failure with preserved EF (HFpEF) with LVEF ≥ 50%, and heart failure with mid-range EF (HFmEF) with LVEF 40–49% [2].

Heart failure with recovered LVEF (HFrecEF) is a newly described clinical entity now emerging as another category of HF, and has its own prognosis and treatment approach [3,4,5]. Several etiologies of cardiomyopathy are known to be reversible following the specific treatment or cessation of those causes including coronary artery disease, myocarditis, peripartum cardiomyopathy, substance or toxin-induced cardiomyopathy, and tachycardia-induced cardiomyopathy. With the advancement of guideline-directed medical therapy (GDMT) and cardiac resynchronization therapy (CRT), recovery of LVEF has become more common [6,7].

From the literature, the definition of HFrecEF varies across studies with the cutoff value for recovery of LVEF ranging from >35% to >50% [8,9]. Heart failure patients with LVEF recovery were shown to have a better quality of life, milder symptoms, and lower rates of rehospitalization and mortality compared to HFrEF patients whose LVEF did not recover [3,10]. However, studies find that patients with HFrecEF are still at increased risk of hospitalization and mortality when compared to patients without HF [3,8,11].

Multiple studies have analyzed predictors of recovering LVEF to better identify patients with an increased chance of LVEF recovery and also determine the prognosis for this patient group [12]. Several predictors of LVEF recovery have been described including younger age, nonischemic origin of cardiomyopathy, and shorter duration of HF [13,14]. The role of gender in HF has been being heavily studied. The prevalence of HFrEF in women is lower than in men, and similarly, women with HFrEF have lower mortality rates when compared to men [15,16]. Some studies reported that female patients have an increased chance of LVEF recovery when compared to men. However, there is a disagreement between published studies regarding this finding. Therefore, we performed a systematic review and meta-analysis to evaluate the association between the female gender and the chance of LVEF recovery in patients with heart failure with reduced ejection fraction.

## 2. Materials and Methods

### 2.1. Search Strategy

Two investigators (AT and ST) independently searched for published studies indexed in the MEDLINE and EMBASE databases from inception to 18 January 2022 using a search strategy including the terms “heart failure” and “recovered ejection fraction” as described in Appendix A. Only full articles in the English language and studies conducted in cohorts were included. A manual search for additional pertinent studies using references from retrieved articles was also completed.

### 2.2. Inclusion and Exclusion Criteria

Studies were eligible if they met the following criteria:(1)Cohort studies (prospective or retrospective), case–control studies, cross-sectional studies, and randomized control trials conducted in HFrEF populations that reported the number of participants who had LVEF recovery, separated by gender.(2)Odds ratio (OR) with 95% confidence interval (CI), or sufficient raw data to perform the calculations were provided. Patients without LVEF recovery were used as controls.

Studies were excluded if they met one of the following criteria:(1)Authors did not report criteria for LVEF recovery, or used parameters other than LVEF to define LVEF recovery;(2)Studies were conducted exclusively in an HF population who had received cardiac resynchronization therapy (CRT);(3)Studies were conducted in patients with left ventricular assist devices or heart transplant recipients;(4)Studies did not report the effect size in their analysis, or there was insufficient data to calculate the effect size.

Study eligibility was independently determined by two investigators (AT and ST) and differences were resolved by mutual consensus. The Newcastle–Ottawa quality assessment scale (NOS) was used to assess each study’s quality. The NOS uses a star system (0 to 9) to evaluate included studies in three domains: recruitment and selection of the participants, similarity and comparability between the groups, and ascertainment of the outcome of interest among cohort and case–control studies. Higher scores represent higher study quality with a maximum score of 9 [17].

### 2.3. Data Extraction

A standardized data collection form was used to obtain the following information from each study: name of the first author, year of publication, country of study, prevalence of ischemic cardiomyopathy, percentage of participants on GDMT, baseline LVEF, LVEF value for LVEF recovery, percentage of participants with LVEF recovery, follow-up duration.

Two investigators (CK and SS) independently performed this data extraction process to ensure accurate data extraction. Any data discrepancy was resolved by reviewing the primary data from the original articles.

### 2.4. Definition

Left ventricular ejection fraction recovery was defined as a measurement of LVEF ≥ 40% with ≥10% absolute improvement in LVEF after prior documented of a decreased LVEF < 40% at baseline [4], or as defined in each study.

### 2.5. Statistical Analysis

We performed a meta-analysis of the included studies using a random-effects model. We pooled the effect size from each study using the generic inverse-variance method of Der Simonian and Laird [18]. The heterogeneity of effect size estimates across these studies was quantified using the test of heterogeneity (I^2^) statistic. The I^2^ statistic ranges in value from 0 to 100% (I^2^ < 25%, low heterogeneity; I ^2^ = 25–50%, moderate heterogeneity; and I^2^ > 50%, substantial heterogeneity) [19]. A sensitivity analysis was performed to assess the influence of the individual studies on the overall results and heterogeneity by the sequential exclusion strategy as described by Patsopoulos et al. [20]. Publication bias was assessed using a funnel plot and Egger’s regression test [21] (*p* < 0.05 was considered significant). All data analyses were performed using STATA SE version 16.0.

## 3. Results

### 3.1. Search Results

Our search strategy yielded 1072 potentially relevant articles (658 articles from EMBASE and 414 articles from MEDLINE). After the exclusion of duplicated articles, 592 articles underwent title and abstract review. Another 530 articles were excluded at this stage since they were not cohort/case–control studies, cross-sectional studies or randomized controlled trials, or were not conducted in the population of interest. This left 62 articles for full-length review. Fifteen studies were further excluded as they were conducted exclusively in patients with CRT and 27 studies were excluded as authors did not report either the gender or the LVEF of participants. Two studies were excluded as the same group of authors used the same database. No additional studies were added through the manual search. Therefore, a total of 18 studies were included in the meta-analysis [3,8,11,14,22,23,24,25,26,27,28,29,30,31,32,33,34,35]. The PRISMA flow diagram is demonstrated in Figure 1.

### 3.2. Description of Included Studies

A total of 18 cohort studies from 2011 to 2021 were included in our meta-analysis with a total population of 12,270 patients (3460, 28.2% female) [3,8,11,14,22,23,24,25,26,27,28,29,30,31,32,33,34,35]. There were 4, 13, and 1 studies that used an LVEF cutoff value of >50%, >40–45%, and >35% for LVEF recovery, respectively. Mean age ranged from 53.6 to 73.1 years old and follow-up time ranged from 6 months to a mean of 67.2 months. Incidence of LVEF recovery ranged from 4.8% to 62.9%. A summary of study characteristics, including prevalence of ischemic etiology, mean LVEF at baseline and percentage of patients on GDMT, is shown in Table 1.

### 3.3. Quality Assessment of Included Studies

The Newcastle–Ottawa quality assessment scale (NOS) is shown in Appendix A.

### 3.4. Meta-Analysis Results

#### 3.4.1. Female and LVEF Recovery

We used unadjusted ORs from the included studies to calculate the pooled effect size to evaluate the association between the female gender and LVEF recovery. Overall, we found that female gender was associated with an increased chance of LVEF recovery (pooled OR = 1.50, 95% CI = 1.21–1.86, *p*-value < 0.001, I^2^ = 74.5%). The forest plot is demonstrated in Figure 2.

We performed a subgroup analysis by the LVEF cutoff definition of LVEF recovery. There were 4, 13, and 1 studies that used LVEF cutoff values of >50%, >40–45%, and >35% for LVEF recovery, respectively. We found that female gender was significantly associated with an increased chance of LVEF recovery when defined by LVEF > 50% (pooled OR = 1.78, 95% CI = 1.45–2.18, *p*-value < 0.001, I^2^ = 0.0%) (Figure 2B), LVEF >40–45% (pooled OR = 1.45, 95% CI = 1.09–1.91, *p*-value = 0.009, I^2^ = 79.2%) (Figure 2A). When defined by LVEF > 35, there was a positive trend towards female gender but did not reach statistical significance (OR = 2.18, 95% CI = 0.94–5.05, *p*-value = 0.06) (Figure 2C).

#### 3.4.2. Sensitivity Analysis

To assess the stability of the results of the meta-analysis, we conducted a sensitivity analysis for each outcome by excluding one study at a time. For each outcome, none of the results were significantly altered, as the results after removing one study at a time were similar to those of the main meta-analysis, indicating that our results were robust.

#### 3.4.3. Publication Bias

To investigate the effect of potential publication bias on the main outcome, we examined a funnel plot generated from the included studies. The vertical axis represents study size (standard error of log OR) while the horizontal axis represents effect size (log OR). From this plot, no bias was observed, as the distribution of studies was symmetrical on both sides of the mean. The funnel plot is demonstrated in Figure 3. Egger’s test was not significant, indicating no small-study effects (*p* = 0.571) [21].

## 4. Discussion

This is the first systematic review and meta-analysis evaluating the effect of female gender on LVEF recovery in patients with HFrEF. The main finding is that female gender is associated with an increased chance of LVEF recovery. The association was significant in the subgroup analysis when recovery of LVEF was defined as LVEF > 40–45% and LVEF > 50%. The association was near significant in the subgroup that defined LVEF recovery as >35%. While it is possible that female gender has less effect in the lower LVEF range, the positive trend suggested that this is more so due to inadequate statistical power.

Among the 18 included studies, 7 studies found that female gender was significantly associated with an increased chance of LVEF recovery [3,8,11,14,26,28,32]. In the other 11 studies, 10 studies reported a non-significant correlation between female gender and LVEF recovery [22,23,24,25,27,29,30] while 1 study found a negative correlation between female gender and LVEF recovery [35]. As shown in Table 1, the studies with non-significant results had a relatively lower number of participants. Thus, it is possible that the lack of statistical significance observed was due to limited power.

Other than female gender, several clinical variables have been speculated to be associated with recovery of LVEF, including nonischemic etiology, absence of left bundle branch block and lower duration of HF [4,7]. We observed from the included studies that HFrecEF populations were less likely to have ischemic etiology. As men generally more commonly have ischemic heart disease earlier in life, it is possible that this is a confounding factor contributing to a lower rate of LVEF recovery [36,37,38]. This is supported by the findings from the study by Swat et al., which was the only study conducted exclusively in patients with nonischemic cardiomyopathy and did not demonstrate a significant association between female gender and LVEF recovery.

The length of reported follow-up time in the included studies varied, ranging from 6 months to a mean of 67 months. The study by Pereira et al. and Bermejo et al. reported a longer follow-up duration at a mean of 60 months. Both studies reported a high incidence of LVEF recovery of 50.7% and 53.1%, respectively. Interestingly, other studies by Basury et al. and Lupon et al. also reported a long follow-up time (mean of 32 and 67 months, respectively) but instead had only 8.4% and 4.8% of participants who exhibited LVEF recovery, respectively. Although Basury et al. and Lupon et al. both used a higher cutoff value for HFrecEF (LVEF of 50% and 45%, respectively), this could suggest that a longer follow-up time does not necessarily correlate with a higher incidence of LVEF recovery.

We found significant heterogeneity in the main analysis (I^2^ = 74.5%) (Figure 2) and the subgroup analysis of LVEF cutoff value >40–45% (I^2^ = 79.2%) (Figure 2A), whereas the subgroup analysis of LVEF cutoff value >50% did not demonstrate any heterogeneity (I^2^ = 0.0%) (Figure 2B). The significant heterogeneity was likely due to different demographics and the striking variance in the incidence of LVEF recovery, which ranged from 4.8% to 62.9%, contributing to different effect sizes.

Selecting a standard and specific cutoff value for HfrecEF is challenging. By choosing a lower LVEF cutoff value for LVEF recovery, more patients would be captured, at the expense of including patients who may not have actually “recovered” the LV systolic function (e.g., LVEF increasing from 30% to 40%). By choosing a higher LVEF cutoff, conversely, the cohort would be enriched by patients more universally considered as having recovered EF (e.g., to LVEF > 50%), at the expense of excluding some patients who have exhibited yet clinically meaningful and substantial recovery but not to a normal range of > 50% (e.g., an improvement in LVEF from 10% to 35%). Nevertheless, a universal definition of HfrecEF would improve and standardize future analyses of HfrecEF by reducing variation in the outcome and endpoint of LVEF recovery.

Our systematic review and meta-analysis have several limitations. First, none of the included studies directly evaluate the association between female gender and LVEF recovery. Thus, all the extracted data were obtained from reported baseline characteristics described in each study without adjusting for confounders, such as age or race. Second, each study used different LVEF measurement techniques, inconsistent definitions for LVEF recovery and had various follow-up times, which likely contributed to the heterogeneity as discussed above. Third, the proportion of patients on GDMT in each study is varied, given that the cohort database that was used in the included studies was obtained from various time periods. Additionally, there is a disparity in the prevalence of ischemic cardiomyopathy and only five studies reported revascularization methods. All of the aforementioned factors likely have a major impact on heterogeneity contributing to the different LVEF recovery rates observed. Finally, despite our best attempts to include only patients without CRT, a modality which can itself promote LVEF recovery and thus confound our results, some of the included studies still included patients with CRT,

## 5. Conclusions

Our meta-analysis demonstrated that female gender is associated with an increased chance of LVEF recovery. Further research that directly evaluates this association with proper confounder adjustment is needed to confirm our findings. A uniform definition of HFrecEF is also imperative to standardize these analyses and findings

## Figures and Tables

**Figure 1 medsci-10-00021-f001:**
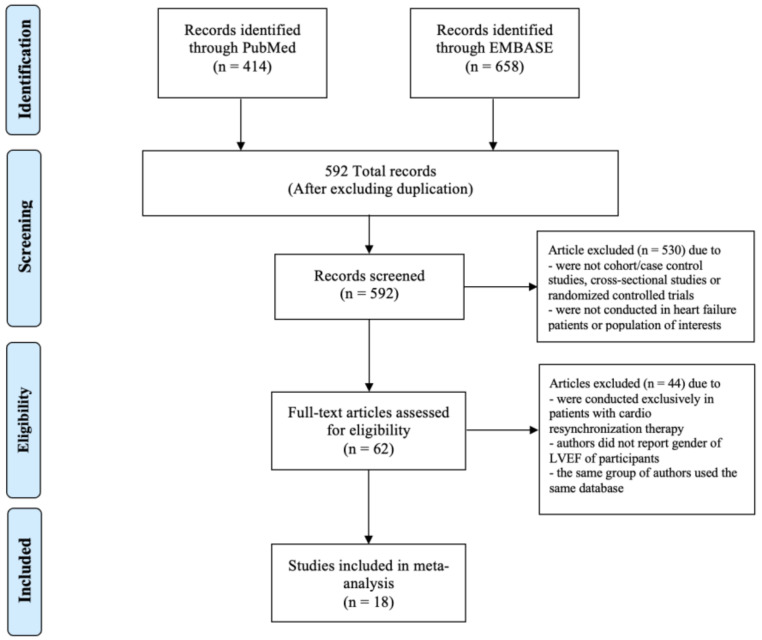
PRISMA flow diagram.

**Figure 2 medsci-10-00021-f002:**
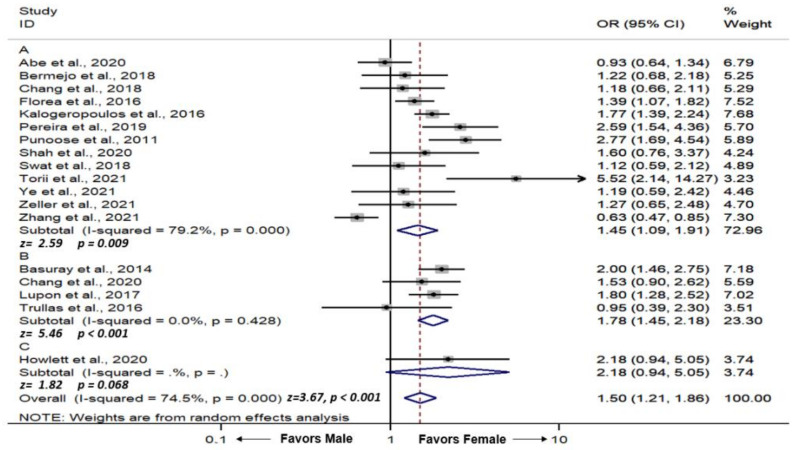
Forest plot demonstrating odds ratio of association between female gender and incidence of left ventricular ejection recovery in patients with heart failure with reduced ejection fraction; (**A**) left ventricular ejection recovery cutoff value of >40–45%, (**B**) left ventricular ejection recovery cutoff value of >50%, (**C**) left ventricular ejection recovery cutoff value of >35%. Square with horizontal line represents OR and 95% CI for each individual study with square size reflecting the statistical weight of the study using the random-effects model. Diamond demonstrates pooled OR and 95% CI for each outcome. Heterogeneity (I^2^) with *p*-value, and pooled effect size (z) with p-value are reported below each of their respective forest plot. CI: Confidence interval; OR: Odds ratio.

**Figure 3 medsci-10-00021-f003:**
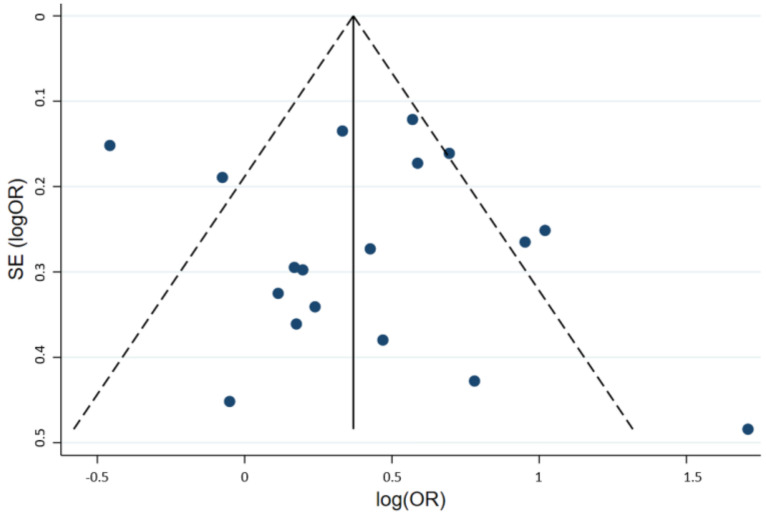
Funnel plot of the included studies. Circles represent included studies. The vertical axis represents study size while the horizontal axis represents effect size. OR: Odds ratio; SE: Standard deviation.

**Table 1 medsci-10-00021-t001:** Characteristics of study participants.

First Author, Year	Country	Total Participants, Female (%)	Mean Age	Definition of HFrEF and LVEF Value for Recovered EF	Participants with Recovered LVEF (n, %)	Ischemic Cardiomyopathy (n, %)		Revascularization Methods (n, %)		Participants on Beta-Blocker (n, %)		Participants on ACEI/ARB (n, %)		Participants on MRA (n, %)		Mean LVEF (%)		Mean Follow-Up Time (Months)
						HFrecEF	HFperEF	HFrecEF	HFperEF	HFrecEF	HFperEF	HFrecEF	HFperEF	HFrecEF	HFperEF	HFrecEF	HFperEF	
Abe, 2020	Japan	567, 28%	64.9 ± 14.6	LVEF < 40% with LVEF ≥ 50% at follow-up via Simpson’s method in 4C view	250, 44%	72, 28.8%	88, 37.4%	PCI 67, 26.8%	PCI 65, 27.7%	230, 92.0%	211, 89.8%	216, 86.4%	187, 79.6%	130, 52.0%	119, 50.6%	33.2 ± 5.7	28.5.2 ± 7.3	40 ± 26.9
Basury, 2014	USA	1699, 32%	56.1 ± 13.9	LVEF < 50% with LVEF ≥ 50% at follow-up	176, 8.4%	29, 16.0%	545, 36.0%	PCI 26, 15%; CABG 18, 10%	PCI 348, 23%; CABG 321, 21%	154, 88.0%	1399, 92.0%	149, 85.0%	1371, 90.0%	35, 20.0%	580, 38.0%	N/A	N/A	32.7 ± 27.4
Bermejo, 2018	Spain	242, 25%	64.4 ± 12.1	LVEF ≤ 40% with LVEF > 40% after 12 months via Simpson’s method	126, 53.1%	25, 19.8%	57, 49.1%	PCI 12, 9.5%; CABG 9, 7.1%	PCI 29, 25%; CABG 14, 12.1%	98, 77.8%	97, 83.6%	113, 90.4%	102, 87.9%	52, 41.3%	63, 54.3%	31.3 ± 6.1	29.1 ± 7.3	60 ± 30
Chang KW, 2018	USA	318, 37.4%	57 ± 12.7	LVEF < 35% with LVEF > 40% after 6 months	59, 18.6%	15, 25.4%	77, 29.7%	n/a	n/a	51, 86.4%	203, 78.4%	52, 88.1%	241, 93.1%	25, 42.4%	115, 44.4%	29.9 ± 3.9	26.7 ± 5.7	6
Chang HY, 2020	Taiwan	437, 25.4%	61.2 ± 14.5	LVEF < 40% with LVEF ≥ 50% after 6 months via biplane Simpson’s method	77, 17.6%	17, 22.1%	160, 44.4%	n/a	n/a	66, 85.7%	300, 83.3%	56, 72.7%	265, 73.6%	47, 61.0%	263, 73.1%	29.3 ± 8	26.5 ± 6.4	18.5 ± 4.5
Florea, 2016	USA	3519, 20.5%	61.9 ± 11	LVEF < 35% with LVEF > 40% after 12 months via biplane Simpson’s method	321, 9.1%	119, 37.0%	1791, 56.0%	n/a	n/a	151, 47.0%	1087, 34.0%	298, 93.0%	2974, 93.0%	13, 4.0%	160, 5.0%	28.7 ± 5.6	25.2 ± 6.2	12
Howlett, 2020	Canada	151, 24.5%	65.2 ± 13.5	LVEF < 35% with LVEF > 35% and absolute increase ≥5% after 12 months via biplane Simpson’s method	95, 62.9%	32, 33.7%	32, 57.1%	PCI 19, 20%, CABG 10, 10.5%	PCI 22, 39.3%, CABG 14, 25%	93, 97.9%	55, 98.2%	85, 89.5%	52, 92.9%	N/A	N/A	44.8 ± 1	24.7 ± 1.2	12
Kalogeropoulos 2016	USA	1700, 37.1%	64.7 ± 15.6	LVEF ≤ 40% with LVEF > 40% at follow-up	350, 20.6%	N/A	N/A	n/a	n/a	310, 88.7%	1180, 87.4%	277, 79.1%	1002, 74.2%	N/A	N/A	25.3 ± 11.2	25.3 ± 11.1	32.3 ± 11.3
Lupon, 2017	Spain	940, 21.6%	65 ± 11.7	LVEF < 45% with LVEF ≥ 45% after 12 months via biplane Simpson’s method	233, 4.8%	82, 35.2%	453, 64.1%	n/a	n/a	217, 93.1%	669, 94.6%	216, 92.7%	669, 94.6%	130, 55.8%	474, 67.0%	31.3 ± 7.7	28.2 ± 7.8	67.2 ± 37.2
Pereira, 2019	Portugal	304, 28.9%	66 ± 14	LVEF < 40% with LVEF ≥ 40% during follow-up	154, 50.7%	39, 25.3%	69, 46.0%	n/a	n/a	152, 97.3%	146, 98.7%	147, 95.5%	142, 94.7%	77, 50.0%	8758.0%	25 ± 8	25.7 ± 8.7	60 ± 39.5
Punnoose, 2011	USA	302, 33.4%	57.4 ± 14.1	LVEF < 40% with LVEF ≥ 40% and absolute increase ≥5% at follow-up	121, 39.8%	21, 17.0%	57, 31.0%	n/a	n/a	99, 82.0%	157, 87.0%	98, 81.0%	146, 81.0%	23, 19.0%	68, 38.0%	25 ± 8	28 ± 12	54 ± 57
Shah, 2020	Saudi Arabia	136, 29.4%	53.6 ± 14	LVEF < 40% with LVEF > 40% with absolute increase ≥10% at follow-up via Simpson’s method	67, 49.2%	9, 13.4%	33, 47.8%	n/a	n/a	61, 91.0%	62, 89.9%	60, 89.6%	60, 86.9%	1, 1.4%	0.0%	26.4 ± 5.75	25.06 ± 7.06	11
Swat, 2018	USA	166, 52.4%	54.3 ± 15.6	LVEF < 40% with LVEF ≥ 40% with absolute increase≥10% within 18 months via longitudinal strain	59, 35.5%	0.0%	0.0%	0.0%	0.0%	40, 68.0%	65, 61.0%	38, 64.0%	64, 60.0%	9, 15.0%	23, 21.0%	26.4 ± 7.4	24.6 ± 8.0	35.9 ± 36
Trullas, 2016	Spain	108, 41.7%	73.1 ± 10.2	LVEF < 50% with LVEF > 50% with absolute increase ≥5% during follow-up	27, 25%	18, 67.0%	46, 57.0%	n/a	n/a	19, 70.0%	61, 75.0%	21, 78.0%	60, 74.0%	6, 22.0%	29, 36.0%	35.3 ± 11.4	31.7 ± 8.3	median 12
Torii, 2021	Japan	100, 30%	67 ± 14	LVEF < 40% with LVEF ≥ 40% with absolute increase ≥10% at 6 months via biplane Simpson’s method	28, 28%	7, 25.0%	24, 33.0%	PCI 7, 25%	PCI 12, 17%	27, 96.0%	66, 92.0%	20, 71.0%	49, 68.0%	18, 64.0%	41, 57.0%	32 ± 4	26 ± 5	24 ± 13
Ye, 2021	China	184, 21.2%	62.1 ± 17.9	LVEF < 40% with LVEF ≥ 40% after 6 months via Simpson’s method in 4C view	88, 21.2%	15, 17.0%	14, 14.6%	n/a	n/a	77, 87.5%	85, 88.5%	71, 80.7%	79, 82.3%	73, 83.0%	87, 90.6%	32.5 ± 6	30.7 ± 6	6
Zeller, 2021	Germany	237, 20.3%	67.4 ± 14.1	LVEF < 40% with LVEF ≥ 40% during follow-up via biplane Simpson’s method	74, 31.2%	29, 39.2%	93, 57.1%	n/a	n/a	64, 86.5%	152, 93.3%	67, 90.5%	139, 85.3%	54, 73.0%	117, 71.8%	31 ± 9	30 ± 9	45.6 ± 28.8
Zhang, 2021	China	1160, 29.8%	61.9 ± 13.3	LVEF < 40% with LVEF ≥ 40% with absolute increase ≥10% after 3 months	284, 26.4%	86, 30.3%	382, 43.6%	n/a	n/a	273, 96.1%	830, 94.8%	233, 82.0%	671, 76.6%	160, 56.3%	626, 71.5%	n/a	n/a	35 ± 20

Abbreviation: 4C: 4 chambers; ACEI: Anigotensin converting esterase inhibitor; ARB: Angiotensin receptor blocker; CABG: Coronary artery bypass graft; EF: Ejection fraction; HFperEF: Heart failure with persistently reduced ejection fraction; HFrecEF: Heart failure with recovered ejection fraction; LVEF: Left ventricular ejection fraction; MRA: Mineralocorticod receptor antagonist; N/A or n/a: Not available; PCI: Percutaneous coronary intervention; USA: United States of America.

## Data Availability

Data is contained within the article or Appendix A.

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
