# Peer review of "Female Gender Is Associated with an Increased Left Ventricular Ejection Fraction Recovery in Patients with Heart Failure with Reduced Ejection Fraction"

_medsci, 2022, doi:10.3390/medsci10020021_

Round 1
Reviewer 1 Report
The work is interesting and clear writing, but there are some places have to be corrected.
- The 3-th affiliation should be corrected.
- Punctuation marks should be added in rows 81-88.
- Font should be corrected in parts 2.3 and 3.1 and dot have to be removed after 2.4 numbers in 104 row
- Could authors to explain in the text the abbreviation I2 in 112 row?
- Table 2 is missed in the text. It should be added.
- The abstract should be corrected according to requirements for authors.
Author Response
The work is interesting and clear writing, but there are some places have to be corrected.
The 3-th affiliation should be corrected.
- Appreciate reviewer’s comment. We corrected the beginning of the 3rd affiliation to “Internal Medicine”
Punctuation marks should be added in rows 81-88.
- Appreciate reviewer’s comment. We added punctuation marks as suggested following each inclusion criteria. We also made correction to numbers of inclusion criteria as well.
Font should be corrected in parts 2.3 and 3.1 and dot have to be removed after 2.4 numbers in 104 row
- Appreciate reviewer’s comment. We corrected font in 2.3 and 3.1 and removed the dot after 2.4.
Could authors to explain in the text the abbreviation I2 in 112 row?
- Appreciate reviewer’s comment. I2 is a standard abbreviation for the test of heterogeneity. We added “test of heterogeneity” in front of I2 in row 112.
Table 2 is missed in the text. It should be added.
- Appreciate reviewer’s comment. We have mentioned table 2 in the text, at row 145
The abstract should be corrected according to requirements for authors.
- Appreciate reviewer’s comment. We made changes in the abstract to followed instruction on the website written as below
“The abstract should be a total of about 200 words maximum. The abstract should be a single paragraph and should follow the style of structured abstracts, but without headings: 1) Background: Place the question addressed in a broad context and highlight the purpose of the study; 2) Methods: Describe briefly the main methods or treatments applied. Include any relevant preregistration numbers, and species and strains of any animals used. 3) Results: Summarize the article's main findings; and 4) Conclusion: Indicate the main conclusions or interpretations. The abstract should be an objective representation of the article: it must not contain results which are not presented and substantiated in the main text and should not exaggerate the main conclusions.”
Our abstract is already less than 200 words and is in a single paragraph format following structured style abstract. Changes we made was we removed the headings per instruction.

Reviewer 2 Report
I propose to correct Table 1 for better understanding. In all columns the same way of representing "number (%)" as
Participants with recovered LVEF n, (%):
250, (44%)
Author Response
Reviewer 2
I propose to correct Table 1 for better understanding. In all columns the same way of representing "number (%)" as
Participants with recovered LVEF n, (%):
250, (44%)
- Appreciate reviewer’s comment. We made changes in table 1 as suggested for ICM and patients on GDMT
Reviewer 3 Report
Kewcharoen et al. evaluated whether female gender is associated with increased left ventricular ejection fraction (LVEF) in heart failure patients with reduced ejection fraction (HFrEF) using systematic review and meta-analysis. They found female gender was associated with an increased chance of LVEF recovery, but nit in EF >35%. However, this paper did not provide enough information to understand the difference between patient characteristics from different trials, which make it difficult to see the follow-up of heart failure.
- The data or information about the anti-angina treatment or PCI for cardiac intervention should be provided also.
- The data shown in the meta-analysis should include the methods to measure the EF, by two-D echo or M mode.
- This analysis should include the racial role to see whether geographic factor may play a role.
- Is there any difference between ischemic cardiomyopathy and non-ischemic cardiomyopathy?
Author Response
Kewcharoen et al. evaluated whether female gender is associated with increased left ventricular ejection fraction (LVEF) in heart failure patients with reduced ejection fraction (HFrEF) using systematic review and meta-analysis. They found female gender was associated with an increased chance of LVEF recovery, but nit in EF >35%. However, this paper did not provide enough information to understand the difference between patient characteristics from different trials, which make it difficult to see the follow-up of heart failure.
The data or information about the anti-angina treatment or PCI for cardiac intervention should be provided also.
- Appreciate reviewer’s suggestion. We added additional column for revascularization method. Only 5 studies reported revascularizations. We added this to limitation on line 254-255.
The data shown in the meta-analysis should include the methods to measure the EF, by two-D echo or M mode.
- Appreciate reviewer’s comment. We added additional details regarding LVEF measurement in the fifth column “Definition of HFrEF and LVEF value for recovered EF” if the method is mentioned in the study. 11 studies described the use of Simpson’s method whereas the rest did not specify the methods. We briefly discussed this in limitation line 250.
This analysis should include the racial role to see whether geographic factor may play a role.
- Appreciate reviewer’s comment. Data on race were not consistently reported across studies. We do not have access to patient-level data of the included studies and thus we were unable to perform analysis separately on race. We briefly discussed this in limitation line 249.
Is there any difference between ischemic cardiomyopathy and non-ischemic cardiomyopathy?
- Appreciate reviewer’s comment. We have column in table 1 describing prevalence of ICM, however, the analysis from each of the included studies were combined outcome between ICM and NICM. There was one study which was conducted solely on NICM (Swat et al.), which showed a non-significant association as shown in the forest plot. Otherwise, we do not have access to patient-level data of the included studies and thus we were unable to perform analysis separately based on this. We briefly discussed this in limitation line 254-257.
Reviewer 4 Report
In this present manuscript, the authors have performed a meta-analysis that suggests that the female gender is associated with an increased chance of LVEF recovery using existing databases of MEDLINE and EMBASE. In general, the manuscript has been crafted meticulously. The rationale and step-by-step experimental approaches have been described properly. As the authors have shown; Among the 18 included studies, 7 studies found that female gender was significantly associated with an increased chance of LVEF recovery, in the other 11 studies, 10 studies reported a non-significant correlation between female gender and LVEF recovery while a 1 study found a negative correlation between female gender and LVEF recovery. Since more than half of the included studies (11) does not show a significantly increased co-relation of increased chance of LVEF recovery in females (although overall has increased LVEF recovery in females), can the authors include a few more additional cohorts (Apart from MEDLINE and EMBASE-If any available) that support their data that female gender is associated with an increased chance of LVEF recovery. Secondly, when forest plots are made these are unadjusted and adjusted values (For example values are adjusted for age, smoking, HDL, LDL, triglyceride level, systolic blood pressure, C-reactive protein level, etc.). Figure 2 forest plot is not age-adjusted I presume. If the forest plot is not age-adjusted, the authors are requested to generate the forest plot with age-adjusted/unadjusted values.
Author Response
Reviewer 4
In this present manuscript, the authors have performed a meta-analysis that suggests that the female gender is associated with an increased chance of LVEF recovery using existing databases of MEDLINE and EMBASE. In general, the manuscript has been crafted meticulously. The rationale and step-by-step experimental approaches have been described properly. As the authors have shown; Among the 18 included studies, 7 studies found that female gender was significantly associated with an increased chance of LVEF recovery, in the other 11 studies, 10 studies reported a non-significant correlation between female gender and LVEF recovery while a 1 study found a negative correlation between female gender and LVEF recovery. Since more than half of the included studies (11) does not show a significantly increased co-relation of increased chance of LVEF recovery in females (although overall has increased LVEF recovery in females)
can the authors include a few more additional cohorts (Apart from MEDLINE and EMBASE-If any available) that support their data that female gender is associated with an increased chance of LVEF recovery.
- Appreciate reviewer suggestion. We additionally searched google scholar, Scopus and Cochrane review. We did not find additional cohort apart from the already included studies.
Secondly, when forest plots are made these are unadjusted and adjusted values (For example values are adjusted for age, smoking, HDL, LDL, triglyceride level, systolic blood pressure, C-reactive protein level, etc.). Figure 2 forest plot is not age-adjusted I presume. If the forest plot is not age-adjusted, the authors are requested to generate the forest plot with age-adjusted/unadjusted values.
- Appreciate reviewer’s comment. The pooled OR from the forest plot were unadjusted. We do not have access to patient-level data of the included studies and thus we were unable to perform analysis adjusted for age. We briefly discussed this in limitation line 249.
Round 2
Reviewer 3 Report
The authors have addressed all of my concerns in the revised manuscript. I have no additional comments.
Reviewer 4 Report
The authors have addressed the comments and concerns as mentioned. Since the authors did not find any additional related cohort data and didn’t have access to patient-level age-related information, it is okay to support their study that the female gender is associated with an increased chance of LVEF recovery with the existing cohort data.